# Motion Responses of a Berthed Tank under Resonance Coupling Effect of Internal Sloshing and Gap Flow

**Tengxiao Wang** [1], **Heng Jin** [1,2,*] , **Mengfan Lou** [1], **Xinyu Wang** [3] **and Yi Liu** [1]

1    Ningbo Institute of Technology, Zhejiang University, Ningbo 315000, China; wangtxnit@163.com (T.W.); jizhilll@163.com (M.L.); liuyilulu@nit.zju.edu.cn (Y.L.)
2    Ningbo Research Institute, Dalian University of Technology, Ningbo 315000, China
3    Shandong Provincial Key Laboratory of Ocean Engineering, Ocean University of China, Qingdao 266100, China; wangxinyu@ouc.edu.cn
*    Correspondence: jinheng@nit.zju.edu.cn

**Abstract:** The growth of global energy transportation has promoted the rapid increase of large-scale LNG (liquefied natural gas) carriers, and concerns around the safety of LNG ships has attracted significant attention. Such a floating structure is affected by the external wave excitation and internal liquid sloshing. The interaction between the structure's motion and the internal sloshing under wave actions may lead to the ship experiencing an unexpected accident. In this research, a hydrodynamic experiment is conducted to investigate the motion responses of a floating tank mooring, both close to and away from a dock. The resonance coupling effect of the internal sloshing and gap flow on the tank's motion is considered. Based on the measured motion trajectory of the floating tank, the stability and safety of the floating tank are estimated. The results show that the sloshing resonance and narrow gap resonance are beneficial to the stability of the ship. This is helpful for controlling the motion of a berthed ship under wave action with a reasonable selection of the gap distance and the liquid level.

**Keywords:** sloshing; narrow-gap resonance; motion response; floating tank

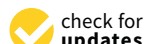



## 1. Introduction

Safety of ocean transportation is the first consideration for marine engineering. Ship stability and its cargo safety depend not only on the design of the ship but also on its interaction with the environmental conditions. The coupling actions of the liquid motion in the ship tank and the external wave loads may cause instability of the ship. Therefore, understanding the motion responses of the liquid cargo ship under the combined actions of multiple loads may be necessary for ship design.

In the past twenty years, the coupling effect between sloshing and ship motion has attracted the attention of many researchers. The coupling actions of the liquid motion in the ship tank and the external wave loads may cause instability of the ship. Faltinsen and Timokha [1] gave a general overview of sloshing in ship tanks. Molin et al. [2] presented a coupled analysis of liquid sloshing and LNG-FPSO based on linear potential theory in the frequency domain. Soon after, a time-domain simulation method with nonlinear viscous sloshing calculation was adopted by Lee et al. [3] and they found that the most pronounced coupling effects are the shift or split of peak-motion frequencies. For understanding the sloshing problem, Brosset et al. [4] introduced a Sloshel joint industry project. A series of researches were conducted based on this project (Bogaert et al. [5]; Lafeber et al. [6]; Maguire J.R. et al. [7]). In addition, Nam et al. [8] provided a hybrid method of impulse-response-function (IRF) and finite difference to solve the ship's motion and nonlinear sloshing problems. After careful experimental validation, they further investigated the sloshing-induced internal forces and their effects on sloshing-induced impact loads. Jiang and Bai [9] numerically investigated the coupling effect of a swaying box and its internal

liquid sloshing under wave actions. The interaction of the floating box and resonance sloshing was visually shown. Moreover, lots of scaled experimental investigations were also used to study the coupling effect. Lee et al. [10] established an experimental test for the ship hull's six degree-of-freedom (6-DoF) motion responses in regular waves for both intact and damaged conditions. Ahn et al. [11] compared the weather side and lee side sloshing impact pressure for LNG cargo tanks under wave action. Similarly, an improved membrane LNG tank was introduction by Xue et al. and its sloshing repression performance was proven [12]. Kim et al. [13] investigated the dynamic pressure on the tank wall and the sloshing-induced pressure of two different tanks were compared. In addition, Xue and Lin. [14] investigated the effect of different liquid baffles and storage vessels' shapes on the suppression of sloshing. They found that the vertical baffle flushing with a free surface is a more effective tool for reducing the impact pressure. In addition, Xue et al. [15] applied a porous material layer to the interior of the liquid tank to obtain optimized parameters in terms of sloshing restriction by varying the porosity, thickness ratio, and average diameter ratio. In addition, the virtual boundary force method was developed to investigate and discuss the efficiency of ring baffles in reducing violent liquid sloshing [16].

In addition to the sloshing inside the ship tank, another phenomenon, which is called narrow gap resonance also affects the stability of the ship. The narrow gap resonance usually appeared when the LNG ships docked next to an FPSO or for transfer operations. The nonlinear motion response may even cause the ship to attack the side structure, although some protection devices have been invented (Metherell and Metherell [17]). Therefore, Kashiwagi et al. [18] and Teigen et al. [19] studied the wave effects related to side-by-side LNG offloading systematically. A variety of investigations on the coupling effect between two floating systems were then carried out (Xiang and Faltinsen [20]; Zhao et al. [21]; Pessoa et al. [22]). Zhao et al. [23,24] reviewed some necessary research topics (motion response, sloshing, gap resonance) for the gap resonance between FLNG-side-by-side offloading safe operations. Once the resonance occurred, the resonant wave height in the narrow gap may reach up to five times the incident wave height (Iwata et al. [25]), which affects the ship's motion significantly. To understand the hydrodynamic characteristic of the resonant fluid in the narrow gap, Moradi et al. [26] investigated wave resonance in the narrow gap between two side-by-side fixed bodies. The body breadth, gap width and draft depth were found to have a significant influence on gap resonance characteristics. Ning et al. [27] considered two barges of different draughts in incident waves. The results showed the wave frequency corresponding to the largest wave amplitude decreased as either barge draught grows. Lu et al. [28] investigated the phenomenon of fluid resonance in narrow gaps between multi-bodies in close proximity under water waves. They found that the resonant frequency moved towards a lower frequency when the gap width increased. Recently, the numerical investigation by Lu et al. [29] revealed the effect of mooring stiffness on the coupling dynamics of the gap resonance and the sway or heave motion of the floating body in regular waves.

While in the harbor, there is a real phenomenon of the superposition of vibrations of two different oscillating systems: sloshing resonance and narrow gap resonance. Different from the previous research, the resonance coupling of sloshing and gap fluid acting on the floating tank were considered together in the present study. A series of experimental tests were performed, considering both resonance behaviors. The aim of the present study is to understand the coupling effect of the inner and outer liquid flow around a float tank. In this paper, the motion responses of a moored floating object under wave action are captured and compared carefully. By adjusting the water levels and the gap distances, the stability of the floating object is discussed under different environmental loads. The present work may have practical significance for improving the safety of vessels transporting liquids.

## 2. Experimental Setup

### 2.1. Experimental Condition

The physical model test of the motion response of the floating tank was carried out in a wave flume with dimensions of 11 m (length) × 0.6 m (breadth) × 1 m (depth). The wave flume is located at the Ningbo Institution of Technology, Zhejiang University. The physical model of the tank was made of acrylic and the dimensions of the model are the length 0.58 m, the width 0.60 m, and the height 0.38 m. The width of the tank along the wave direction was presented by $W$. In the present experiment, the sides of the tank model were in contact with the flume and the 0.015 m gap was filled by four universal wheels on both sides, which ensures that the movement of the floating tank can be constrained into a two-dimensional plane. The floating tank was designed to meet the draught requirement of the liquid cargo ship, so that the coupling resonance of the sloshing and gap flow can occur simultaneously. The draught depth ($d$) was set to 0.14 m through a reasonable ballast evenly arranged within the tank.

Three wave probes were installed around the floating tank (shown in Figure 1) to measure the wave height. Two displacement sensors, M1 and M2, were fixed on the dock and outer frame, respectively, and their flexible sides were anchored on the same location of the floating tank for measuring the motion of the tank. In addition, an angle sensor was placed at the center top of the floating tank for capturing the rotation movement of the tank.

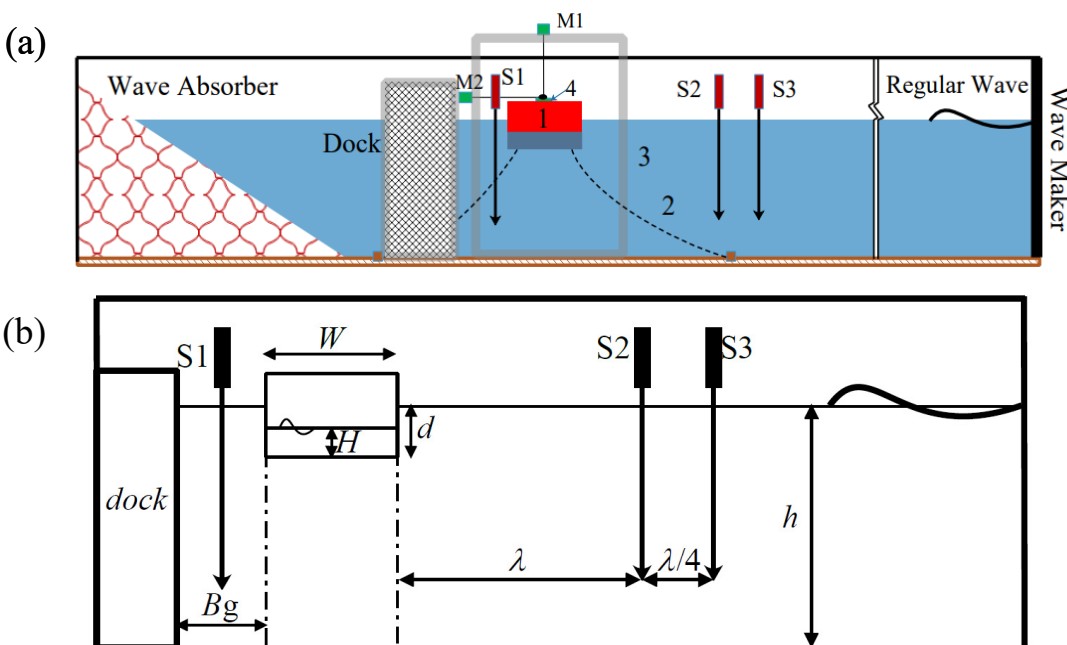

**Figure 1.** (**a**) The layout of the experimental wave tank; (1) floating box, (2) mooring line, (3) outer fixed frame, (4) inclination sensor, (M1/2) Displacement sensors, (S1/2/3) Wave probes; (**b**) Parameter setup of the experimental model, $\lambda$ is wave length.

### 2.2. Problem Description

The motion responses and hydrodynamic characteristics of the floating tank under wave actions in the open sea were first tested. The water level in the wave flume is $h = 0.40$ m. The incident wave with a period of $T = 1$ s and wave height of $H_0 = 0.04$ m.

The natural period of the tank was estimated according to Equation (1), which was derived by Faltinsen and Timokha [30], where $T_n$ is the natural period of the floating tank. Varying filling levels in the floating tank $H$ (0.05 m to 0.15 m, with $\Delta H = 0.2$ m) were then

tested in the present experiment. Among them, the $H$ = 0.07 m is the resonant condition of the floating tank.

$$T_n = \frac{2\pi}{\sqrt{\frac{n\pi g}{W}\tanh(\frac{n\pi H}{W})}}, n = 1, 2, 3, \ldots, \tag{1}$$

The motion responses and hydrodynamic characteristics of a berthed floating tank under wave action were then studied. A vertical dock is considered in this condition. Therefore, a narrow gap appears behind the floating tank.

$$T_g = 2\pi\sqrt{\frac{\frac{B_g W}{h-d} + d}{g}}. \tag{2}$$

The frequency of the narrow gap was estimated by Equation (2), $T_g$ is the natural period in the gap, $B_g$ denotes the gap size between tank and dock model. The liquid levels in the floating tank $H$ = 0.07 m, 0.11 m, 0.13 m and 0.15 m and gap widths $B_g$ (0.06 m and 0.09 m) were taken into account in this study. The gap resonance appeared at $B_g$ = 0.06 m with $T_g$ = 1.06 s. The test arrangements are shown in Table 1 (in detail, cases 1:1-6 studied motion response of floating tank in the open sea and cases 2:1-6 focused on the floating tank in the berthing condition).

**Table 1.** Parameters arrangement of the test cases.

| Cases | $H$ (m) | $B_g$ (m) | $T_n$ (s) | Sloshing Resonance | $T_g$ (s) | Narrow Gap Resonance | Coupled Resonance |
|-------|---------|-----------|-----------|-------------------|-----------|----------------------|-------------------|
| 1-1 | 0.05 | o | 1.17 | o | o | o | o |
| 1-2 | 0.07 | o | 1.01 | | | o | o | o |
| 1-3 | 0.09 | o | 0.92 | o | o | o | o |
| 1-4 | 0.11 | o | 0.86 | o | o | o | o |
| 1-5 | 0.13 | o | 0.81 | o | o | o | o |
| 1-6 | 0.15 | o | 0.79 | o | o | o | o |
| 2-1 | 0.13 | 0.06 | 0.81 | o | 1.06 | - | o |
| 2-2 | 0.13 | 0.09 | 0.81 | o | 1.19 | o | o |
| 2-3 | 0.15 | 0.09 | 0.79 | o | 1.19 | o | o |
| 2-4 | 0.15 | 0.06 | 0.79 | o | 1.06 | - | o |
| 2-5 | 0.07 | 0.06 | 1.01 | | | 1.06 | - | + |
| 2-6 | 0.07 | 0.09 | 1.01 | | | 1.19 | o | o |

"-" is narrow gap resonance, "|" is sloshing resonance, "+" is coupled resonance, "o" is non-resonance.

In this paper, the duration of each test is set to 100 s. But the stable intermediate data (30–80 s) was intercepted for the analysis below. The displacement amplitude of each direction was obtained and the results were expressed as AmpX (surge), AmpY (heave) and AmpR (pitch).

## 3. Results and Discussion

### 3.1. Motion Response of Floating Tank in Open Sea

Figure 2a gives the motion response of the floating liquid tank in the open sea. The heave response was reduced effectively when resonant sloshing occurred (Table 1, case 1-2) and the surge motion was insensitive with variations in the liquid sloshing status. Further observation of the heave response in Figure 2b indicates that the amplitude of the heave motion of the floating tank with the sloshing resonance approximates 50.8% of that without sloshing action. (Table 1, case 1-1). The effects of the sloshing resonance on suppressing the motion of the floating tank are obvious.

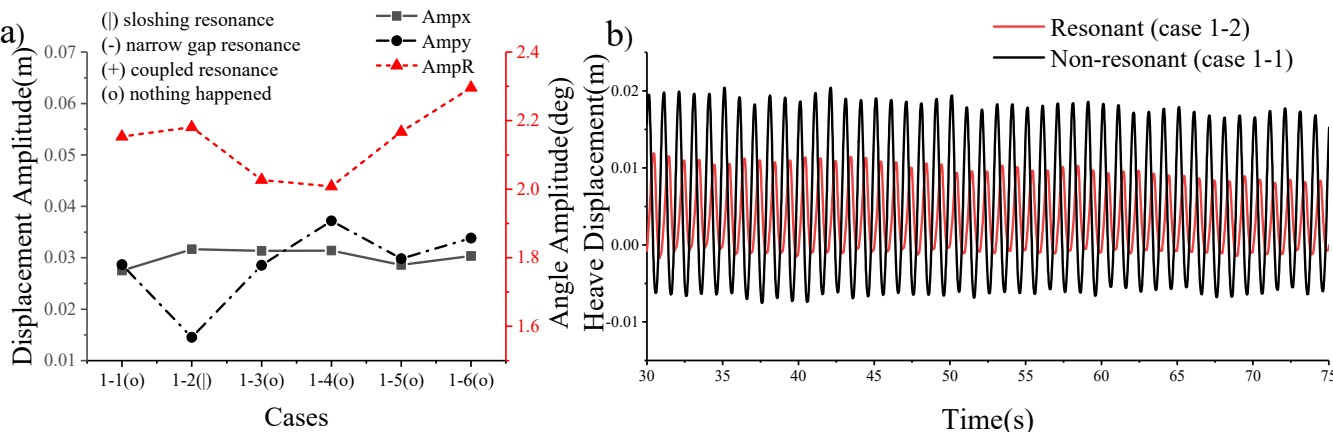

**Figure 2.** (**a**) Motion response of floating tank in the open sea; (**b**) Time histories of swing motion of floating tank under resonant and non-resonant conditions.

### 3.2. Motion Response of the Berthed Floating Tank

The motion responses are compared between the open sea (Table 1, cases 1: 1-6) and berthing conditions (Table 1, cases 2: 1-6). Figure 3 shows that the liquid motion in the ship's tank and the narrow gap has little effect on the surge motion of the floating tank, which is relatively steady. The heave motion is significantly reduced when the resonant sloshing occurred (Table 1, cases 2-5 and 2-6). However, when sloshing resonance occurred (Table 1, case 1-2, cases 2-5 and 2-6), the restriction on the heave motion of the floating tank is significant. The heave motion can be reduced by about 40%. The pitch of the floating tank fluctuated irregularly with the variation of the liquid level in the tank, which may be caused by the non-linear motion of the liquid flow. Under berthing conditions, the pitch of the floating tank is reduced compared to the open sea. Furthermore, Table 2 gives the dispersion coefficients of the motion response to express the fluctuation characteristics of the movement, which was the ratio of standard dispersion and the mean of each amplitude under two different environmental loads. The results illustrate that the surge of the floating tank is significantly affected by the variation of environmental loads. This can also be confirmed from the results in Figure 3 that the heave of the floating tank is reduced due to the coupling resonance.

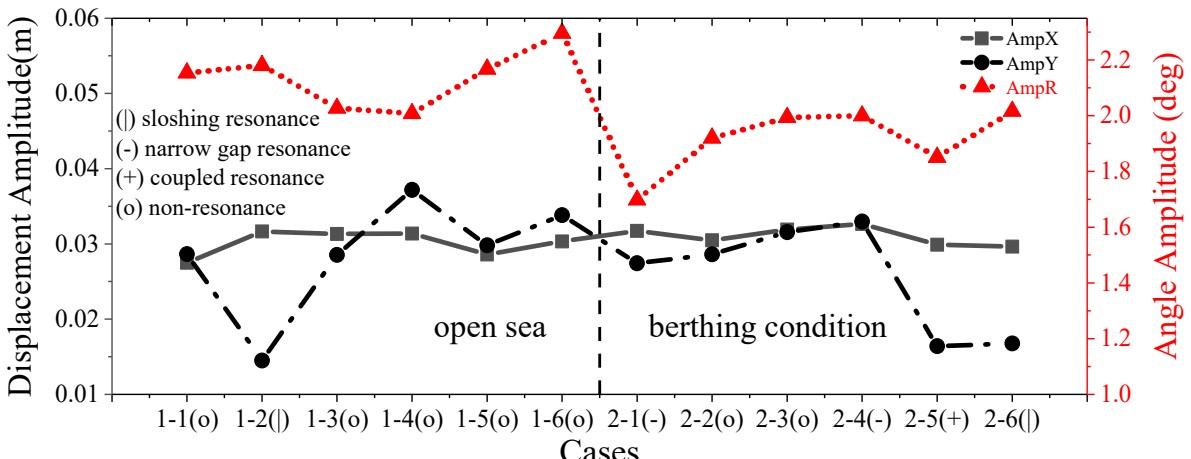

**Figure 3.** Motion response of floating tank under different conditions.

**Table 2.** Dispersion coefficients of motion response.

| Deviation Coefficient | Open Sea | Berthing Condition |
|---|---|---|
| AmpX | 0.052036 | 0.035876 |
| AmpY | 0.246216 | 0.259292 |
| AmpR | 0.045661 | 0.058453 |

The reflection and transmission coefficients of cases are depicted in Figure 4. The incident and reflect wave were separated by using Goda's method [31] and then the reflection coefficients were calculated by the reflected wave height divided by the incident wave height. The transmission coefficients are the ratio of the wave height in the narrow gap to the incident wave height. They indicate that changes in environmental loads have little effect on the reflection coefficient. In the berthing condition, it is clear that the transmission of the coefficient increases significantly due to the appearance of a narrow gap. Concerning Figure 3, the pitch motion of the floating tank has an overall decline under berthing conditions, which may be influenced by the fluid motion in the narrow gap. Meanwhile, three obvious resonant actions (Table 1, cases 2-1, 2-4 and 2-5) occur when $B_g$ = 0.06 m; they lead to a significant elevation of wave height in the leeside of the floating tank. A corresponding phenomenon is observed in Figure 3, in which the narrow gap resonance can reduce the pitch amplitude of the floating tank.

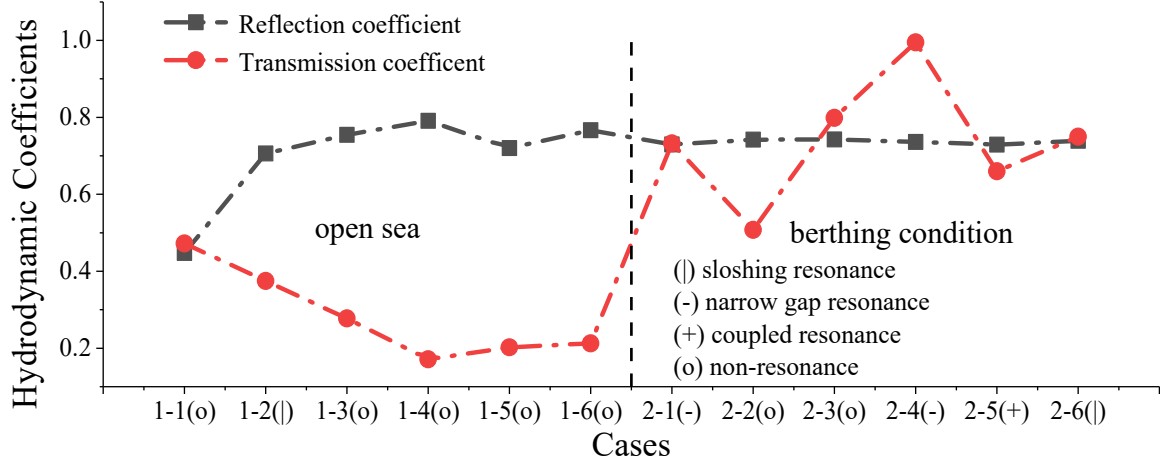

**Figure 4.** Transmission and reflection coefficients of the floating tank.

### 3.3. Motion Response of Berthed Floating Tank

To further understand the motion response of the berthed liquid cargo vessel under wave action, the frequency analysis of the motion response and wave height in the narrow gap are provided in Figure 5. Figure 5a,b show that the amplitude of the tank motion is significantly reduced when the sloshing resonance occurs. The higher order wave frequencies emerge as double and triple times frequencies in the floating tank backwash. Unlike a single narrow gap or sloshing resonance, when coupled resonance occurs, the transmitted waves towards the floating tank backwash have a wider distribution at higher order frequencies. A corresponding phenomenon is shown in the motion response diagram that the coupled resonance leads to a reduction in the rotation amplitude of the floating tank. However, the actual wave frequency is still at around the first order incident wave frequency. In this case, in the frequency domain of the wave height in Figure 5a, part of the wave frequency lies near the narrow gap resonant period, $T_g$ = 1.06 s. Meanwhile, the resonance leads to a dispersion of wave energy and a significantly lower wave amplitude than in other cases.

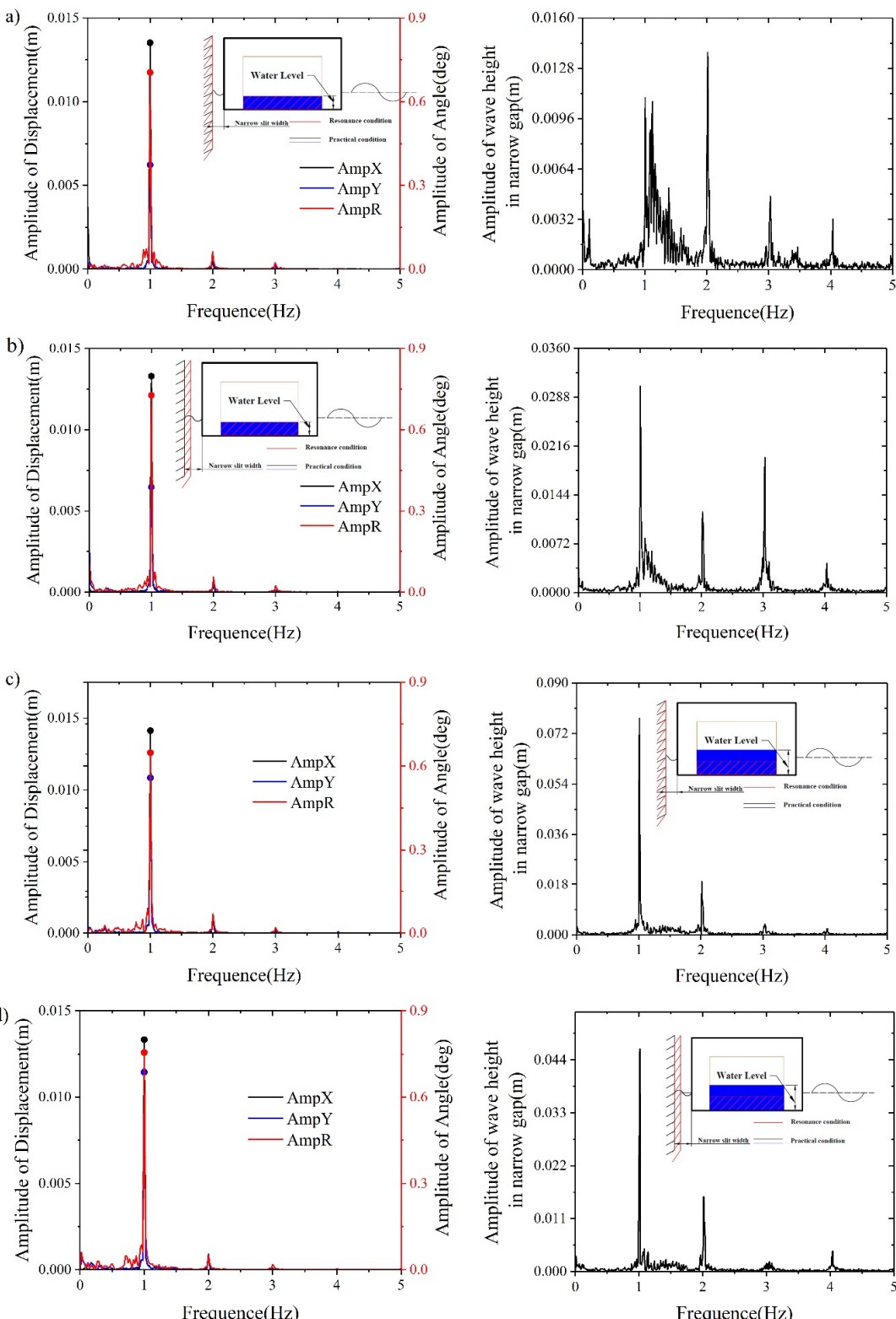

**Figure 5.** Motion response of a berthed floating tank. (**a**) The double resonance of sloshing and the narrow gap in Case 2-5(+); (**b**) the sloshing resonance in working Case 2-6(|); (**c**) narrow gap resonance in Case 2-1(-); (**d**) the absence of resonance in Case 2-2(o).

### 3.4. Collision Test of the Floating Tank with Dock

Finally, the distance ($D$) between the floating tank and the dock was discussed (Figure 6). The distance was calculated by Equation (3), which is affected by the initial gap, the translation in wave direction and the horizontal component of rotation. Figure 7 illustrates the real-time dimensionless distance ($(D - B_g)/B_g$) of the berthing tank. It can be seen from Figure 7 that the floating tank did not collide with the dock under the present experimental series. This is consistent with the result in Figure 3. Because the maximum translation was 0.033 and the rotation angle was 2.16°, and therefore the minimum distance was $B_g - |X| + W/2(1 - \cos(R)) = 0.028$ m. In addition, when the sloshing resonance occurred individually (Table 1, case 2-6), the surge of the floating tank was restricted and the trajectory was compressed. Meanwhile, the floating tank had a smaller distance amplitude. When the sloshing resonance and narrow gap resonance occurred simultaneously (Table 1, case 2-5), the motion trajectory of the tank was more concentrated and, therefore, beneficial to the stability of the ship in berthing situations.

$$D(t) = B_g + X(t) + \frac{W}{2}(1 - \cos R(t)). \tag{3}$$

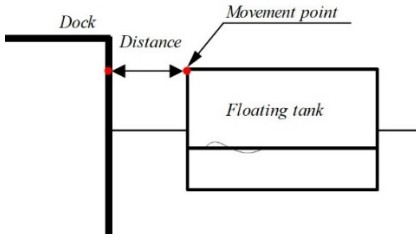

**Figure 6.** Distance between floating tank and dock.

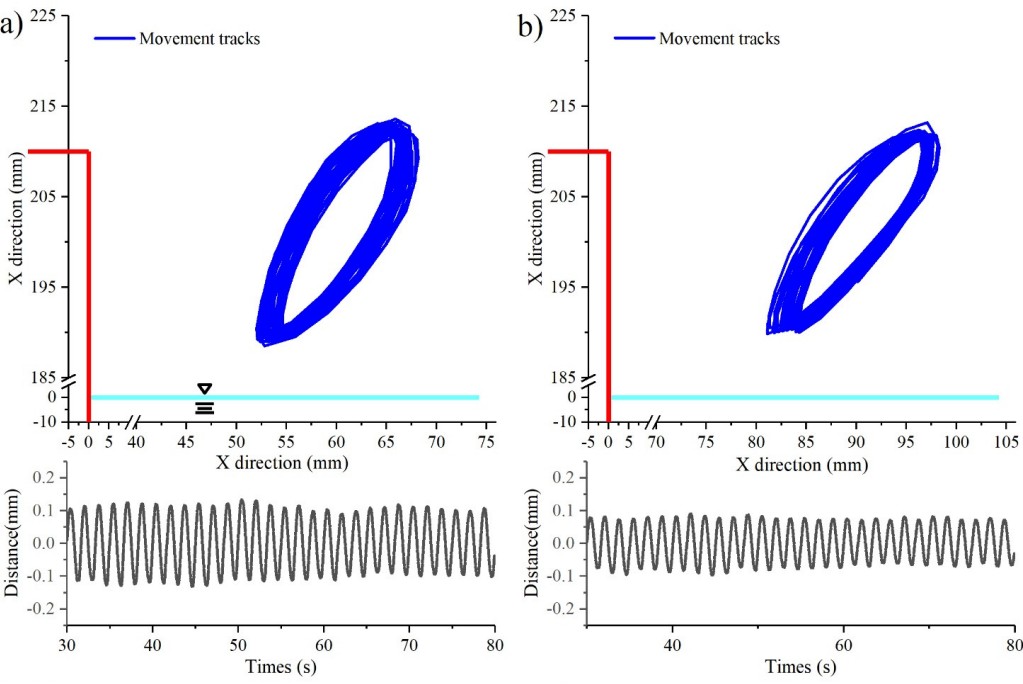

**Figure 7.** *Cont.*

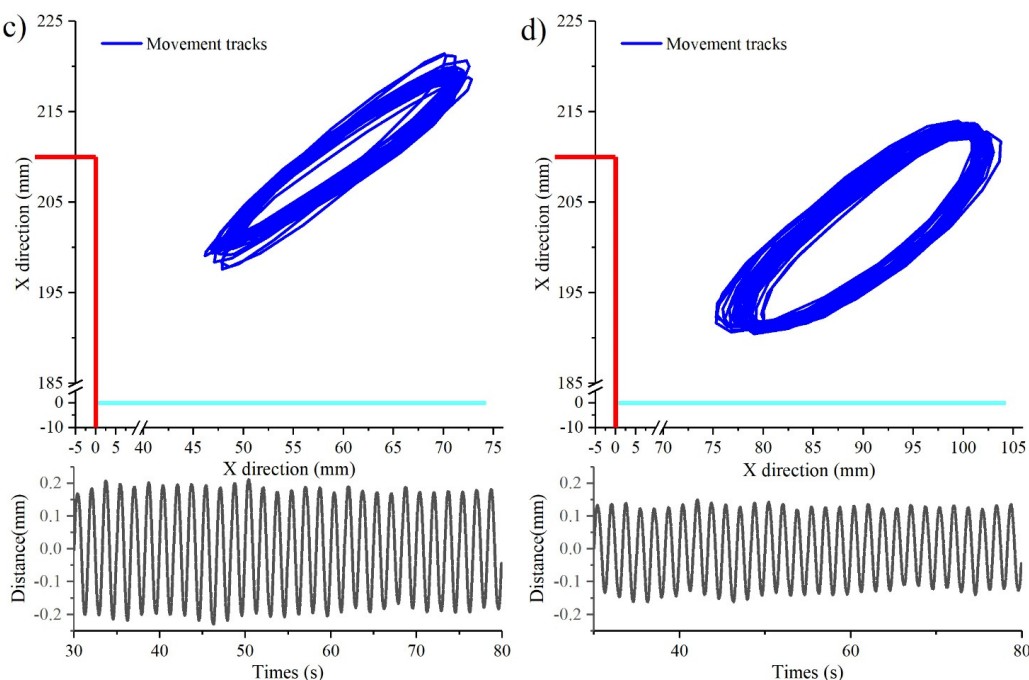

**Figure 7.** Movement tracks and distance between floating tank and dock of four different resonance conditions. (**a**) The coupled resonance of sloshing and the narrow gap in case 2-5; (**b**) the sloshing resonance in case 2-6; (**c**) the narrow gap resonance in case 2-1; (**d**) the non-resonance in case 2-2. The red and blue lines represent the position of the dock and the still water level surface, respectively.

## 4. Conclusions

The coupled effect of sloshing and narrow gap resonance on the motion response of the liquid tank was investigated based on the experimental test in this research. The water level of the floating tank and narrow gap width was adjusted as variables and the motion response of the floating tank was analyzed. The main conclusions are listed as follows:

1.  A reasonable liquid tank design can reduce the amplitude of the heave response under wave action through tank resonance and improve the stability of the ship at anchor;
2.  When narrow gap resonance occurs, the wave height between the narrow gap increases significantly. The sloshing resonance has a beneficial effect on the stability of ships;
3.  The surge motion response of the floating tank is minimally affected when resonance occurs;
4.  The occurrence of the coupling resonance may be beneficial to the stability of the vessel when berthing;
5.  Based on the experimental cases in this paper, no collisions occurred in all cases and the resonance of sloshing and narrow gap contributed to avoiding a collision of the floating body with the dock.

**Author Contributions:** Writing—original draft, T.W.; Writing—review and editing, H.J., Y.L. and X.W.; Experimental testing and data processing—T.W. and M.L. All authors have read and agreed to the published version of the manuscript.

**Funding:** This research was funded by the National Natural Science Foundation of China, grant number 52001276, the Public Welfare Foundation of Ningbo, grant number 2021S094, the Natural Science Foundation of Zhejiang, grant number LQ19E090005, Shandong Province Natural Science Foundation Youth Branch, grant number ZR2021QE188 and Project of Department of Education of Zhejiang Y202044011.

**Conflicts of Interest:** The authors declare no conflict of interest.

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
