# Peer review of "Motion Responses of a Berthed Tank under Resonance Coupling Effect of Internal Sloshing and Gap Flow"

_water, doi:10.3390/w13243625_

Round 1

Reviewer 1 Report

In this research, the coupling effect of liquid tank sloshing and narrow gap resonance on a floating tank is investigated by using the experimental test. The wave-induced motion responses of floating tank under open sea and berthing condition are considered. The heave, surge and pitch motions in a 2D plane are analyzed for the above two conditions. Also, the interaction between wave propagation and structure response is revealed. Besides, collision analysis of the floating tank with a dock is conducted. The study is interesting and it can be applicable in the harbor industry. This study can be published by improving the text and addressing the following comments. 1 It is expected to have much more recent references in the Introduction. 2 In the introduction, the authors need to highlight more, what are the differences between this study and others? What is the novelty? 3 In Line 86. A 14 cm draft is selected. Please explain that. 4 In section 2.2, the authors described the resonant frequency of sloshing and narrow gap. While the definition of natural period Tn in Table 1 is missing. Besides, it may be more appropriate to use a consistent period or frequency to describe resonance conditions in this research. 5 The result of equation 2 is the natural frequency of the narrow gap. The definition of natural frequency Tg in Line 112 is not correct. It should be ωg. Besides, please keep all parameters in the article consistent. 6 What’s the relationship between Figures 3 and 4. More discussions are needed. 7 The discussion about the results in Table 2 is not clear. The present conclusion is too simple. 8 In Figure 7, it may be better to use wave direction for the x-axis and vertical direction for the y-axis. 9 The process to calculate the distance between the floating tank and the dock should be stated. 10. There are many grammar and spelling errors, please correct them.

Author Response

Dear reviewer,

Thanks for your kind comments. Your advice is very helpful for improving the quality of our work. With your help, we hope the present version of the manuscript can meet the requirement of the journal.

Best wishes

All authors of this article

Reviewer 2 Report

The themes and objectives of the paper are interesting.

However, it took me a long while to understand the purpose of the paper.

The paper analyzes the coupling between fluid oscillations in a floating tank anchored near a wharf and water level oscillations in the space between the tank and the wharf induced by surface waves at a given frequency. The authors studied the effect of the liquid level in a floating anchored tank on its oscillations by looking for the liquid level at which resonant conditions occur. They also studied how the distance between the anchored tank and the wharf affects the amplitude of the water level in that area. They looked for the distance at which resonance would occur. Both processes were excited by generated surface waves of the same period (same frequency). Thus, resonance conditions at the same frequency were sought for both phenomena.

The authors attempted to induce vibrations of two systems wanting to show the result of the coupling of their vibrations under resonance conditions.

They presented results in four cases:

  1. vibration far from the resonance conditions of both systems.
  2. resonance of the first system - second system out of resonance
  3. resonance of the second system - first system out of resonance
  4. both systems in resonance.

The title of the paper is difficult to understand.

I would suggest changing it to something else.

For example, "Experimental study of the resonance coupling effect of a sloshing liquid in a tank anchored a short distance from the wharf in the presence of surface waves".

The text is written very synthetically and briefly which makes it difficult to understand and analyze its content.

Relating Table 1 to subsequent figures is somewhat hampered by the lack of direct description of the cases in it.

The text: "The experimental setup is shown in Table 1." does not seem to be sufficient. It is understandable for the authors but not for the reader.

No description of the cases considered. They are numbered, but it is not clear what they represent. No reference to table 1.

It took me a long time to link the cases considered to the contents of this table.

Typing errors => 2.2 Problom Description

Definition of the reflection coefficient. The measurement system has three free surface height sensors. It should be understood that the reflection and transmission coefficients are calculated from the measurements taken by the two sensors between the wave generator and the hull.

Unintelligible sentence. "However, when sloshing resonance occurred (Cases 2-5 and 2-6), the reduction of the floating tank is significant".Reduction of what?

How was the wave dispersion calculated?

It seems that the readability of Figures 2, 3, and 4 could be improved by adding text information on them about the existence of resonance and in which system it occurs. This information is in Table 1 but should be repeated in the figures.

In the abstract, the authors note the negative effects of fluid resonance in a moving tank. But in the paper, they use resonance in a moving tank to indicate its utility in damping tank vibrations. The problem is that although the tank (vessel) is oscillating with a smaller amplitude, nothing is known about what is happening to the liquid in its inner tank. And the liquid in the tank oscillates in resonance conditions, which means large amplitudes of motion.

The authors probably have information about the form and amplitude of the oscillation of the liquid in the transparent tank, but they do not present this information.

The paper proves that the distance of the ship from the quay can affect the movements of the ship's hull. That the liquid level in the ship's tank can be the reason for the appearance of resonant vibrations in the liquid tank. That the simultaneous appearance of liquid resonance in the ship's tank and in the space outside the ship, between the ship and the wharf, can, at certain distances of the ship from the wharf, lead to a reduction in the amplitude of the ship's motions. And this is a very important conclusion.

If there is such a phenomenon, it is likely that there may be a reverse phenomenon where the amplitude of the ship's motions increases.

Usually, the addition of an additional counter-phase vibrating system will reduce the amplitude of vibration at the resonant frequency but resonances at lower and higher frequencies will occur. This is probably evident in Fig. 5 in the right column for the 2.5 -double resonance case.

The second column in Fig. 5. does it show the amplitudes recorded by the S1 sensor?

The paper is very interesting. It presents a very real example of the superposition of vibrations of two different oscillating systems and their effects. It may have practical significance in improving the safety of vessels transporting liquids.

However, understanding the authors' ideas requires a lot of effort from the reader. The paper contains all the necessary information but I would suggest a broader and clearer presentation.

The paper can be published after a slight expansion of its descriptive part and perhaps supplementing the drawings with suggested additions to facilitate their interpretation.

Author Response

Dear review,

Thank you for your time to review our article and also glad that you are interested in our work. We would like to express our sincere thanks to you for your comments on improving the quality of our manuscript. Your advice and kindly suggestions are very helpful to us. We have tried our best to answer all of your questions and revised the manuscript. We hope the present version of the manuscript can meet you and the journal's requirements.

Best wishes,

Heng Jin, Tengxiao Wang, Mengfan Lou and Xinyu Wang 

Round 2

Reviewer 1 Report

It can be accepted in its current form. I do not have further comments on this manuscript.